# Relation Modeling of Factors Influencing Life Satisfaction and Adaptation of Korean Older Adults in Long-Term Care Facilities

**DOI:** 10.3390/ijerph17010317

**Published:** 2020-01-02

**Authors:** Soonyoung Park, Sohyune R. Sok

**Affiliations:** 1Department of Nursing, Shingyeong University, Hwasung-si, Gyeonggi-do 18274, Korea; qjt10@daum.net; 2College of Nursing Science, Kyung Hee University, Seoul 02447, Korea

**Keywords:** aged, adaptation, life satisfaction, long-term care facility

## Abstract

This study aimed to model and examine the relationship between the factors influencing the adaptation ability and life satisfaction of the elderly people living in long-term care facilities. This study used a cross-sectional descriptive design and relation prediction modeling. Participants were 229 older adults over 65 years old, who had lived for more than six months in the long-term care facilities of the Gyeonggi and Gyeongsang provinces, South Korea. The model construction was based on the Ecological model developed by Lawton (1982). The data were included demographics, physical health status, emotional health status, self-efficacy, and social support. The analysis of collected data was done by using the SPSS 22.0 and AMOS 22.0 programs. The model fit index for the modified model was χ^2^ = 15.561, χ^2^/df = 2.223, GFI = 0.980, AGFI = 0.920, NFI = 0.967, CFI = 0.970, and RMRS = 0.018, RMSEA = 0.021. Life satisfaction was influenced by the factors of adaptation, depression, anxiety, friend support, self-efficacy, and staff support. In addition, adaptation was affected by the factors of staff support, depression, anxiety, and friend support. This study suggests that life satisfaction and adaptation for Korean elderly in long-term facilities were primarily influenced by of the factors of anxiety, depression, friend support, and staff support. In the nursing practice, nurses need to pay attention to these factors to improve the life satisfaction and adaptation ability of Korean elderly in long-term facilities.

## 1. Introduction

In South Korea, which is one of the fastest aging countries in the world due to rapid industrialization and the changes in the family structure, it is difficult for other family members to take care of an elderly family member [1]. Home care has been taken for granted in the types of large families with multiple children that were promoted by traditional society. However, the family unit has gradually been reduced to a nuclear family with a small number of children [2,3,4]. In today’s society, women, who traditionally were the main family care managers, are increasingly participating in the workface and have less time to carry out family support tasks, thus contributing to families being increasingly unable to look after elderly people at home [1,2]. Consequently, the number of elderly people entering elderly care facilities, where they are to spend the rest of their lives, is increasing [5].

However, the elderly people who leave their homes unexpectedly and enter an institution with a new environment develop a sense of isolation, mistrust, and confusion, and think that they have lost a part of themselves as a result [6]. They experience a new environment where the unfamiliar and strange approach of the nursing staff makes it an inhospitable facility [3,6]. This may lead to deterioration of their physical function and daily activity, sleep disorder, anxiety, chaos, depression, hopelessness, loneliness, isolation, memory impairment, suicide, etc., and may cause serious maladaptation to and lowered life satisfaction in the facility [7,8].

Adaptation to the facility refers to the acceptance of a facility as one’s home and a positive perception of life in the facility by reversing one’s sense of rejection in the new place of residence and changing one’s cognition, feelings, and lifestyle [1]. Life satisfaction in the facility means the positive feeling of happiness that one experiences upon adapting to life in the facility and accepting the facility as one’s home [9], and can be a potential indicator of successful aging and psychological adjustment [10]. The degrees of adaptation and life satisfaction of the elderly people entering elderly care facilities at the present time, when the number of elderly people living in such facilities is increasing, are essential factors for healthy old age at such a facility [11].

As long-term care services started being available for the elderly, the factors influencing their decision to enter the facility included socio-demographic characteristics (e.g., age, gender, material status, education level, and income level) and family support characteristics (e.g., family living and family burden) [2,12]. Other factors affecting this decision included cognitive and behavioral characteristics, activities of daily living, the degree of illness, and health status [2,12,13]. In addition, the responses and behaviors of friends and neighbors near the nursing home during the preparation phase of the decision-making process have an impact on the choice to enter a facility [3,12]. The decision to enter the facility, the degree to which they want to enter during the admission process, the pre-planning for entering the facility, and the preparation of a strategy related to life in the facility affect the adaptation ability and life satisfaction of the elderly after entering a facility [1,4,14]. In previous studies, it was found that the factors influencing the adaptation of such elderly people to life in a facility included the changes in their physical and mental symptoms [11,15], self-efficacy [13], self-esteem [16], anxiety [17], depression [13], social support [16], and the facility admission situation [3]. Also in such studies, it was found that the factors influencing the life satisfaction of the elderly people living in a facility included their health status, subjective health, positive thinking [18], anxiety [19], depression [13], self-efficacy [1], self-esteem [7], social support [20], and financial support from the government [21]. Factors that may influence the experience of the elderly people who are living in a facility include social support as part of the environment in the facility [11,20,22]. Social support includes family, friends, and neighbors outside the family, whereas the facility is composed of colleagues, friends, and employees. The emotional support and help of family members, friends, and employees of the facility have an effect on the life satisfaction of the elderly who are living in the facility [4,11,16]. Stronger support and satisfaction from human resources, particularly the employees in a facility, can improve life satisfaction in that facility [11,20,22]. As such, social support plays an important role in predicting the life satisfaction of the elderly in a facility.

Most of the previous studies were about fragmentary influencing factors such as physical and emotional factors, social and psychological factors, and social system factors. Therefore, it is necessary to investigate the structural causal relationship for adaptation to and life satisfaction in a facility, which is comprehensively affected by physical, emotional, social, and psychological factors, as well as by social support.

The conceptual framework of this study was established as shown in Figure 1, based on the ecological model of Lawton [23] and the relevant-literature review. In this study, based on the constitutive factors of the ecological model by Lawton [23], the personal characteristics consisted of the physical health state, the emotional health state including anxiety and depression, and the social and psychological factor of self-efficacy. The environmental characteristics consisted of the subdomains of social support, including support from one’s family, one’s fellow residents in the facility, and the facility staff. The adaptation behavior involved the ability to adapt to the elderly care facility, and the emotional response involved life satisfaction in the facility. Based on the conceptual framework of this study, a hypothetical model was established as presented in Figure 2. The aims of this study were to examine the general characteristics of study participants, and to analyze and examine the relationship between the factors influencing the adaptation and life satisfaction of the elderly people living in elderly care facilities.

## 2. Material and Methods

### 2.1. Study Design and Participants

This study used a cross-sectional descriptive design as a model construction study for verifying the fit of the final model. The participants of this study were elderly people aged 65 years or older who had been living in the elderly care facilities in the Gyeonggi and Gyeongsang provinces for more than six months. Inclusion criteria were elderly people who understood the purpose of the study, had the cognitive ability to respond, and signed the informed consent form for study participation to signify their voluntary consent to participate in the study. Exclusion criteria were elderly people who had two or more types of pharmacological treatments. The participants were selected through random sampling while considering the convenience of data collection. In terms of the sample size, more than 200 persons were required to use the maximum likelihood (ML) estimation method [24].

In this study, 258 elderly people were approached and asked to participate, and eight did not want to take part. Then, taking the dropout rate into consideration, 250 questionnaires were distributed, and excluding the 21 questionnaires that were not completed, 229 were finally collected and analyzed (Figure 3). 

### 2.2. Measures

The Short Portable Mental Status Questionnaire (SPMSQ), a mental state test developed by Pfeiffer [25], was revised, translated, and reverse translated by two nursing professors. This scale was used in order to measure the degree of cognitive ability of the participants. It consisted of a total of 10 questions with correct (one point) or incorrect (zero point) responses, and the score range was 0 to 10 points. The higher the score of the respondent, the higher their level of cognitive ability. In this study, the reliability was Cronbach’s α = 0.963. Only eight points or above were included in this study.

A general characteristics survey for the study participants was developed by researchers and it consisted of eight items in total: gender, age, education, religion, spouse, main guardian, frequency of keeping in touch with friends or family, and frequency of receiving support. Characteristics related to entering a long-term care facility included the ‘facility placement decision maker’, ‘main motivating factor for entering a long-term care facility’, ‘length of stay’, and ‘payment for entering a long-term care facility’.

The Korean Health Status Measure for the Elderly (KHSME) was developed by Shin et al. [26] and was revised by Kim [27]. This scale included physical health status, anxiety, and depression for Korean elderly people. The physical health status scale, including physical pain in the scale, was used to measure the degree of physical health status of the participants. The anxiety and depression scales as emotional health status in this scale were used to measure the degrees of anxiety and depression of the participants. It consisted of a total of 27 questions (physical health status: 14 items, anxiety: five items, depression: eight items) using a 5-point Likert scale. The range of scores for physical health status was 14 to 70 points, the range for anxiety was 5 to 25 points, and the range for depression was 8 to 40 points. The higher the score of the respondent, the better their levels of physical health status, anxiety, and depression. The reliabilities of these measuresments in this study were Cronbach’s α = 0.884, 0.853, and 0.889, respectively.

The General Self-Efficacy Scale (GSES) developed by Sherer et al. [28] was translated and revised by Oh [29] into a Korean version. This scale was used to measure the degree of self-efficacy of the participants. It consisted of a total of 17 questions using a 5-point Likert scale, and the score range was 17 to 85 points. The higher the score of the respondent, the higher their level of self-efficacy. The reliability of this measure in this study was Cronbach’s α = 0.958.

The Multidimensional Scale of Perceived Social Support (MSPSS) developed by Zimet, Dahlem, Zimet, and Farley [30] was translated by Lee [31] into a Korean MSPSS. This scale was used to measure the degree of social support obtained by the participants. This scale included family support, friends support, and special support provided by employees of a facility. It consisted of a total of 12 questions using a 5-point Likert scale, and a score range of 12 to 60 points. The higher the score of the respondent, the higher their level of social support. The reliability of this measure in this study was Cronbach’s α = 0.855.

The adaptation to an elderly care facility scale was developed by Lee [32]. It consisted of a total of 23 questions using a 5-point Likert scale, and the score range was 13 to 65 points. The higher the score of the respondent, the higher their level of adaption and self-efficacy. The reliability of the scale in this study was Cronbach’s α = 0.857.

The Memorial University of Newfoundland of Scale for Happiness (MUNSH) developed by Stones and Kozma [33] was translated and revised by Yang [34] into a Korean MUNSH. This scale was used to measure the degree of life satisfaction of the participants. This scale included emotion (five items), experience (10 items), and relative satisfaction (five items). It consisted of a total of 20 questions using a 5-point Likert scale, and the score range was 20 to 100 points. The higher the score of the respondent, the higher their level of life satisfaction. The reliability of the scale in this study was Cronbach’s α = 0.845.

### 2.3. Procedures

A preliminary survey was conducted from 20 March to 10 April 2017 to determine the questionnaire measurement tools that would be used in this study. A preliminary survey was conducted on 50 elderly people from elderly care facilities with less than 100 persons. Exploratory factor analysis, confirmatory factor analysis, and a reliability test for preliminary items including general characteristics, admission-related characteristics, and cognitive ability items were conducted, and 99 items were determined as the final questionnaire items. The main survey included a preliminary survey of 50 elderly people.

The data collection period of the survey was from April to July 2017. To select the study subjects, random sampling, a nonprobability sampling method, was used for elderly people aged 65 years or older living in elderly care facilities. The data were in principle collected through a self-administered questionnaire survey. When subjects had difficulty in filling out the questionnaire, the researcher read out the questions and wrote down the subject’s answers for them. Only 15 subjects had difficulties in filling out the questionnaires. After the purpose of the study was explained to the subjects, a written informed consent was obtained from those among them who showed a willingness to voluntarily participate in the study. After measuring the subjects’ cognitive abilities using SPMSQ, the general characteristics of the individual subjects were identified, and then a questionnaire survey was conducted on their adaptation to and life satisfaction in the facility.

### 2.4. Statistical Analysis

The collected data were analyzed using SPSS 22.0 and AMOS 22.0. The frequency, percentage, and descriptive statistics were analyzed for the subjects’ general and admission-related characteristics. Additionally, the reliability of the measured variables was tested, and the Pearson correlation coefficient was verified to check the multicollinearity of the measured variables. The mean, standard deviation, skewness, and kurtosis were examined to confirm the normality of the major variables constituting the study model. Finally, the fit of the study model constructed based on literature review was evaluated, and the relationships among the variables were analyzed. Then path analysis was conducted to verify the applicability of the hypothetical model. In the path analysis, the confounding factors were not considered, and only an estimation of the sample size was presented based on statistical methodological research and prior studies [7,24]. In the study, multi-item Likert scales could be treated as interval scales for a path analysis based on statistical methodological research and prior studies [1,7,24,35].

### 2.5. Ethical Considerations

After ethical consideration, the Institutional Review Board of K hospital in Seoul, Korea approved this study (KHSIRB-16-023(NA)). Participants were informed that they were voluntarily taking part in this study and could withdraw from the study at any time. Participants were also informed of the confidentiality of the data. Researchers obtained completed written consent forms from the study participants.

## 3. Results

### 3.1. General Characteristics of Study Participants and Characteristics Related to Enter a Long-Term Care Facility

The general characteristics of study participants and characteristics related to entering a long-term care facility are shown in Table 1 and Table 2. Among the study participants, there were more females (67.2%) than males (32.8%). The mean age of the subjects was 82.26 years, and the 80-years-old-or-older age group was the largest age group at 72.5%. Regarding marital status, 30.1% had a spouse, and 69.9% were widowed or divorced. The person(s) who usually provided the elderly people staying in the elderly care facility with the most support was usually a family member or family members. As for the facility-admission-related characteristics of the elderly people, the decision of admission to the facility was made by their children in 63.3% of the cases. The reason for admission to the facility was a health issue due to a disease in 72.1% of the cases. The health issue due to a disease rationale accounted for more than half of the cases. The most common admission period was one year to below three years (47.2%) (Table 1 and Table 2).

### 3.2. Descriptive Statistics of Measured Variables

Descriptive statistics of measured variables are presented in Table 3. The Z-scores of the skewness and kurtosis of all the variables used in the study did not exceed the threshold (±1.96) at the 0.05 significance level, confirming that the normal distribution condition was satisfied (Table 3). The reliability of the measured variables was analyzed based on the Cronbach’s α value, which evaluates the internal consistency of the questions. The Cronbach’s α value was 0.70 or higher for all of the variables, indicating that there were no problems in the reliability of the measured variables. With regard to the multicollinearity of the correlation coefficient matrix, the correlation coefficient between the variables was less than 0.90, showing that the possibility of the presence of multicollinearity was low (Table 3).

### 3.3. Standardized Direct, Indirect, Total Effects for the Modified Final Model

The hypothetical model was based on the collected data and the literature review. The goodness of fit in the hypothetical model was characterized by an χ^2^ statistic of 60.553, χ^2^/df of 7.569, GFI (Goodness-of-Fit Index) of 0.935, AGFI (Adjusted Goodness-of-Fit Index) of 0.715, NFI (Normed Fit Index) of 0.924, CFI (Comparative Fit Index) of 0.928, RMRS (Root Mean-Squared Residual) of 0.018, and RMSEA (Root Mean Squared Error of Approximation) of 0.080. The goodness of fit in the modified model improved significantly after adding the direct effect from adaptation ability to life satisfaction.

The modified final model in this study is shown in Figure 4. The goodness of fit was characterized by a χ^2^ statistic of 15.561, χ^2^/df of 2.223, GFI (Goodness-of-Fit Index) of 0.980, AGFI (Adjusted Goodness-of-Fit Index) of 0.920, NFI (Normed Fit Index) of 0.967, CFI (Comparative Fit Index) of 0.970, RMRS (Root Mean-Squared Residual) of 0.018, and RMSEA (Root Mean Squared Error of Approximation) of 0.021. These values were found to satisfy the recommended levels, and the modified model was confirmed as the final model. In the final model, the χ^2^ statistic was 15.561, and the significance probability was *p =*.261, which were appropriate overall. In both the hypothetical model and the modified final model, the variables influencing the adaptation included staff support, depression, friend support, and anxiety, in descending order. The listed variables explained 54.1% of the adaptation level (Adjusted R^2^ 0.541, *p <* 0.001). The physical health state, self-efficacy, and family support, however, were not significant. In the modified final model, after adding the relationship between the adaptation level and life satisfaction, the variables influencing life satisfaction included adaptation level, depression, anxiety, friend support, self-efficacy, and staff support, in descending order. The listed variables explained 72.6% of the life satisfaction (Adjusted R^2^ 0.726, *p <* 0.001). On the other hand, the family support and physical health state were not significant (Figure 4). An increase in the R2 of the modified model as compared to that of the hypothetical model (0.726 − 0.541 = 0.185) was statistically significant at the 0.01 level. The addition of the path from the adaptation to the life satisfaction variables yielded a substantial improvement in the model’s goodness-of-fit.

Standardized direct, indirect, and total effects for the modified model are shown in Table 4. In the process of modifying the model, the adaptation between exogenous variables such as depression, anxiety, friend support, staff support, and life satisfaction showed a partial mediating effect by changing the factor load (path coefficient). The significance of the total effect, direct effect, and indirect effect was also verified. Among the exogenous variables in this study, depression had the largest direct effect on life satisfaction, followed by anxiety, friend support, and staff support (Table 4).

## 4. Discussion

The factors influencing elderly people’s adaptation to the elderly care facility where they were residing included anxiety, depression, and support from their fellow residents in the facility and from the facility staff. The factors influencing elderly people’s life satisfaction in the facility included anxiety, depression, self-efficacy, support from their fellow residents in the facility and from the facility staff, and their adaptation to the facility. As a result, the factors influencing the elderly people’s adaptation to and life satisfaction in the elderly care facility where they were residing were their emotional health state (anxiety and depression) and social support (from their fellow residents in the facility and from the facility staff), suggesting that emotional support, support from one’s fellow residents in the facility, and support from the facility staff are necessary for successful adaptation and for maintaining high life satisfaction. Self-efficacy affected the life satisfaction of elderly subjects living in elderly care facilities, and therefore to maintain high life satisfaction in a elderly care facility, successful adaptation to the facility is needed.

In the present study, the better the friend and staff support were, the better the adaptation to and the higher the life satisfaction in an elderly care facility for elderly persons. The results of the present study are consistent with those of the studies whose results showed that the higher the social support was, the more positive the adaptation to a facility [15,35] and the better the life satisfaction [20,31] for elderly residents. In addition, this is consistent with the result that the stronger the support from one’s fellow residents in the facility and from the facility staff, the higher the elderly resident’s adaptation ability to and life satisfaction in a facility. It is also consistent with the result that social support affected life satisfaction in an elderly facility, and staff support wielded the greatest influence [20,36]. This emphasizes the role of one’s friends in old age, and implies that support from one’s friends has a greater influence on elderly people’s adaptation to and life satisfaction in an elderly care facility than family support, indicating the importance of the other elderly people a resident lives with in the facility [37]. Until recently, the elderly people in South Korea regarded filial piety and family support as the most important forms of social support because of established norms [3,16,20]. In addition, the decision to enter a facility, including health care, is often considered to be a more important decision for their children than for the elderly people themselves [3,19]. However, once the elderly people start living in a facility, they are able to continue their life there, influenced by the staff and their fellow residents in the facility, and they form a new emotional support group that increases their overall quality of life.

As the results of the final model show, the better the adaptation to the elderly care facility was, the more their life satisfaction increased. This result of the present study is the same as that of another study on elderly people living in elderly care facilities [9,11]. In contrast, there was a report suggesting that life satisfaction a the facility causes an improved adaptation ability to that facility rather than the other way around [21]. Our results suggest that the life satisfaction of elderly people facilitates adaptation to a new society or environment, while adaptation can also be seen as an important factor for predicting life satisfaction. Therefore, elderly people can experience life satisfaction while adapting to an unfamiliar environment and can enjoy a happy life during old age.

For the successful adaptation and high life satisfaction of the elderly people living in elderly care facilities, management of anxiety and depression and support from their fellow residents in the facility and from the facility staff are necessary. In particular, these findings highlight the importance of recognizing mental health (e.g., anxiety and depression) in the elderly people who are living in facilities. To lower the anxiety and depression of the elderly people, the facility staff can visit them before admission and provide them with various information related to the facility, such as on the facility’s strengths and weaknesses, locations, daily schedules, meals, services, family visits, and discharge procedures as well as the characteristics of the other elderly people living in the facility and of the facility staff [38]. In addition, when elderly people enter an elderly care facility, it is important to allow them to bring items that are familiar to them, to understand their past life based on the information obtained through conversations or interviews with them and their families, and to provide them with care based on a sufficient understanding of their needs. For elderly people who have started living in elderly care facilities, it is necessary to develop and provide leisure activity programs that can offer them social support through connections with the inside and outside of the facility, while providing them with programs for reducing their anxiety and depression. Elderly people face difficulties when they start living in an elderly care facility and need to establish new relationships with strangers. As such, the facility staff should actively intervene and work to enable them to develop and maintain good relationships with their fellow residents and the facility staff, and to raise their intimacy levels. Ciccone et al.’s study [38] reported that elderly people’s health and self-management abilities developed overwhelmingly positive outcomes through a strong “partnership” between the care manager and the patient, as well as through effective collaboration between the physician and the care manager. A variety of facility staff education programs must be developed to improve the staff’s relationship-building skills and professionalism.

As the number of the elderly people in the facilities is increasing in South Korea, long-term care facilities are also trying to apply and conduct person-centered care by using the Korean version of the Resident Assessment Instrument-Facility (RAI-FC) assessment tool within the first 14 days after a new resident enters their nursing facility and at least once a year for all facility members [5]. Person-centered care that is based on the findings from this study is deemed to be necessary in order to improve the adaptation ability and life satisfaction of the elderly people in the facility.

A longitudinal study was deemed to be necessary for future research in order to accurately investigate the overall structural causal relationship for the adaptation and life satisfaction of elderly people in a facility that is comprehensively affected by various complex dimensions, including physical, emotional, social, and psychological factors. Also, further qualitative research is needed to identify and deeply understand the inner world of elderly people with regard to their own thoughts on their adaptation to and life satisfaction in the elderly care facility where they are residing. Based on this, it is necessary to conduct experimental research for practical and effective intervention development and application, to improve the adaptation and life satisfaction of the elderly people living in care facilities.

This study has some limitations. First, the residences in this study were limited to four elderly care facilities in the Gyeonggi and Gyeongsang provinces, and random sampling was used. As such, care should be taken when generalizing the results. Second, the data collected in this study was, in principle, based on a self-administered questionnaire survey with a high number of questions. Thus, there is a limitation in the individuals’ accurate expression of their circumstances.

## 5. Conclusions

In conclusion, the appropriateness of a path model in which multiple factors influencing the adaptation and life satisfaction of elderly people living in elderly care facilities could explain their adaptation and life satisfaction was confirmed. Through this study, it was found that anxiety and depression affect the adaptation and life satisfaction of elderly people living in elderly care facilities, and that the support from their fellow residents in the facility and from the facility staff is important. Based on the results of this study, it can be concluded that strengthening the validated factors will lead to the successful adaptation and high life satisfaction of elderly people living in elderly care facilities.

## Figures and Tables

**Figure 1 ijerph-17-00317-f001:**
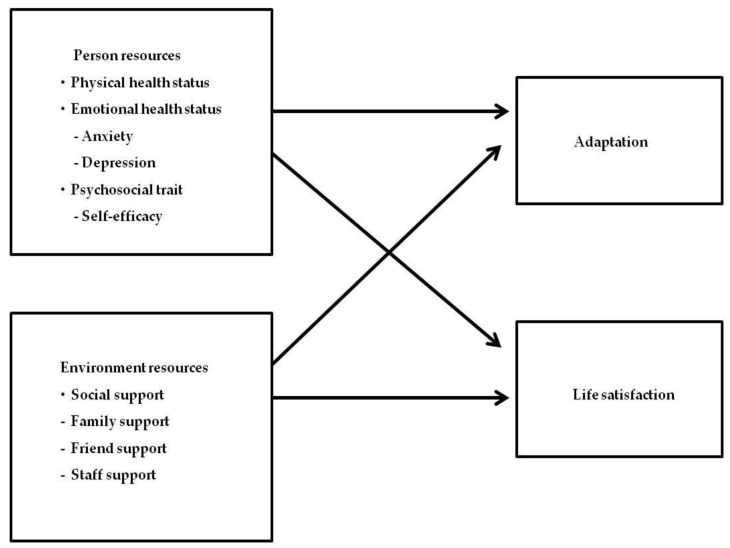
The Conceptual framework.

**Figure 2 ijerph-17-00317-f002:**
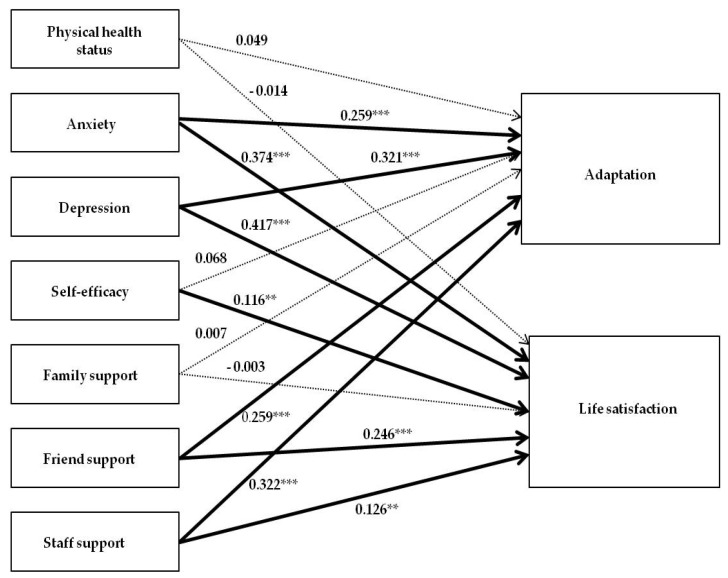
The hypothetical model.

**Figure 3 ijerph-17-00317-f003:**
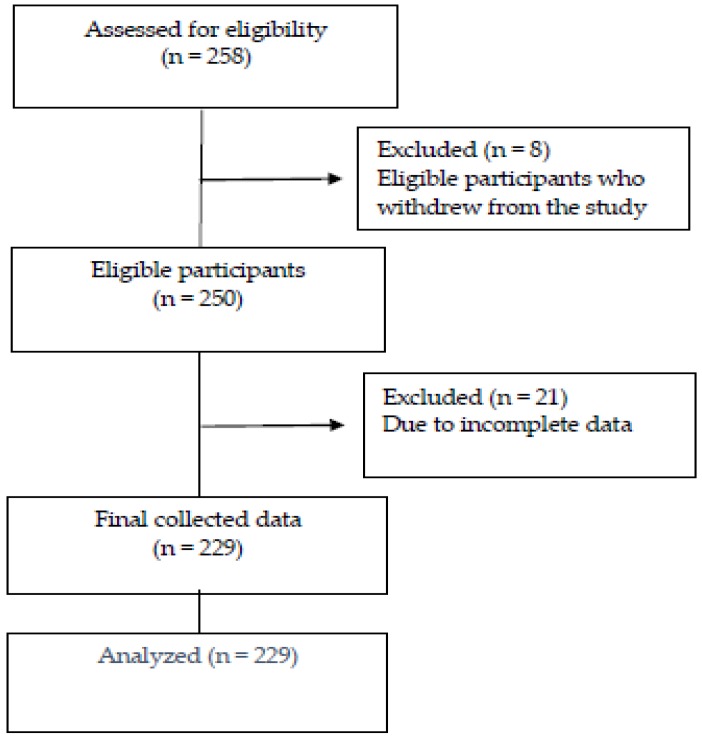
Flow chart of the study.

**Figure 4 ijerph-17-00317-f004:**
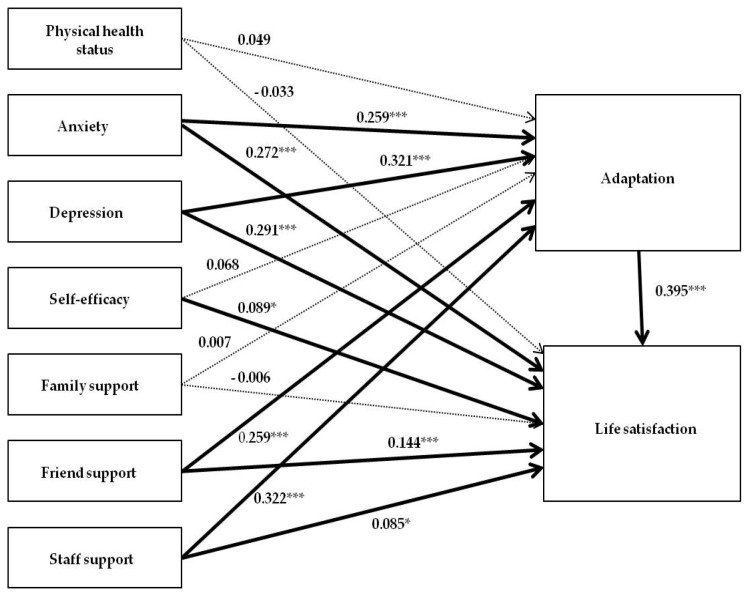
The modified final model.

**Table 1 ijerph-17-00317-t001:** General characteristics of study participants (*n* = 229).

Characteristic	Variables	*n* (%)
Gender	Female	154 (67.2)
	Male	75 (32.8)
Age (years)	65–69	14 (6.1)
	70–74	20 (8.7)
	75–79	29 (12.7)
	80–84	66 (28.8)
	85–89	74 (32.3)
	≥90	26 (11.4)
Education	None	71 (31.0)
	Elementary school	104 (45.4)
	Middle school	25 (10.9)
	High school	24 (10.5)
	College or higher	5 (2.2)
Religion	Protestant Christianity	83 (36.2)
	Buddhism	84 (36.7)
	Roman Catholicism	13 (5.7)
	None	45 (19.7)
	Etc.	4 (1.7)
Spouse	Yes	69 (30.1)
	No	160 (69.9)
Main guardian	Spouse	12 (5.2)
	Child(ren)	176 (76.9)
	Brother or sister	5 (2.2)
	Family relatives	1 (0.4)
	None	35 (15.3)
Keeping in touch frequency (number/month)	0	45 (19.7)
	1–7	144 (62.8)
	8–15	28 (12.3)
	16–30	12 (5.2)
Most frequent source of support	Family	170 (74.2)
	Family relatives	2 (0.9)
	Neighborhood	11 (4.8)
	Friends	1 (0.4)
	None or Pastor, Nun, Monk	45 (19.7)

**Table 2 ijerph-17-00317-t002:** Characteristics related to entering a long-term care facility (*n* = 229).

Characteristic	Categories	*n* (%)
Facility placement decision marker	Self (participant)	62 (27.1)
	Spouse	10 (4.4)
	Child(ren)	145 (63.3)
	Daughter-in-law	2 (0.9)
	Brother or sister	1 (0.4)
	Pastor or Government local administrative staff	9 (3.9)
Main motivating factor for entering a long-term care facility	Comfort during older age	38 (16.6)
	No caregiver	25 (10.9)
	Heath problem (disease)	165 (72.1)
	None	1 (0.4)
Length of stay (years)	½–<1	77 (33.6)
	1–<3	108 (47.2)
	3–<5	21 (9.2)
	≥5	23 (10.0)
Payment for entering a long-term care facility	Free	13 (5.7)
	Payment required	216 (94.3)

**Table 3 ijerph-17-00317-t003:** Descriptive statistics of measured variables.

Variables	M †	SD ‡	Skewness	Kurtosis
Physical health status	2.370	0.712	0.765	0.355
Anxiety	3.232	0.639	–0.833	0.921
Depression	2.998	0.589	–0.216	–0.546
Self-efficacy	3.168	0.587	–0.306	–0.553
Family support	3.400	0.838	–0.082	–0.254
Friend support	2.883	0.645	0.319	0.198
Staff support	3.943	0.562	0.226	0.382
Adaptation	3.300	0.348	–0.304	0.561
Life satisfaction	3.337	0.355	0.071	–0.407

† M = Mean; ‡ SD = Standard deviation.

**Table 4 ijerph-17-00317-t004:** Standardized direct, indirect, and total effects for the modified final model.

Exogenous Variables	Endogenous Variables	SDE †	SIE ‡	STE §	SMC¶
Anxiety	Adaptation	0.259 ***		0.259 ***	0.541 ***
Depression		0.321 ***		0.321 ***	
Self-efficacy		0.068		0.068	
Friend support		0.259 ***		0.259 ***	
Staff support		0.322 ***		0.322 ***	
Anxiety	Life satisfaction	0.272 ***	0.102 *	0.374 ***	0.726 ***
Depression		0.291 ***	0.127 ***	0.417 ***	
Self-efficacy		0.089 *			
Friend support		0.144 ***	0.102 ***	0.246 ***	
Staff support		0.085 *	0.127 ***	0.212 ***	
Adaptation		0.395 ***		0.395 ***	

† SDE = Standardized direct effect; ‡ SIE = Standardized indirect effect; § STE = Standardized total effect; ¶ SMC = Squared multiple correlation; * *p* < 0.05, ** *p* < 0.01, *** *p* < 0.001.

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
