# Peer review of "Relation Modeling of Factors Influencing Life Satisfaction and Adaptation of Korean Older Adults in Long-Term Care Facilities"

_ijerph, 2020, doi:10.3390/ijerph17010317_

Round 1

Reviewer 1 Report

This paper reports a study of factors influencing the adaptation and life satisfaction of the elderly people living in long-term care facilities in South Korea.

Overall, the procedures followed in the study appear to be sound.  However, the statistical analysis and its description can be improved by the following.

First, the table of descriptive statistics need to include the sample means and standard deviations of the measured outcome variables.

Second, you need to make clear that the regression models you estimate and report are conventional multiple regression models with the (Likert) adaptation and happiness scales as the outcome variables.  And you need to cite statistical methodological research  or prior studies that conclude that such multi-item Likert scales can be treated as interval scales (for a conventional multiple regression analysis), that is, that such regression analyses generally lead to inferences that are similar to those that treat the scales as ordered categorical scales with intervals between integers that are variable.

Third, on first use in the text, you need to spell out the names of the model fit statistics in the text on first use, e.g., GFI, AGFI, etc.  I know their meaning, but this is a general social science journal intended for readers who may not be statisticians.  Among these model fit statistics, you also need to list R-squared,

Fourth, in Figure 2, you report the regression model estimates for the  "modified final model".  Okay, but it would be good also to report first the path diagram of the estimated conceptual model -- without the link from Adaptation to Life Satisfaction and then report the estimates for this final model, thus showing the model contribution of the Adaptation to Life Satisfaction link.

Fifth, Table 3 needs work:  You need to make clear which of the coefficients relate to which outcome variable.

Author Response

Revision Comments

Reviewer 1

Comments and Suggestions for Authors

This paper reports a study of factors influencing the adaptation and life satisfaction of the elderly people living in long-term care facilities in South Korea.

Overall, the procedures followed in the study appear to be sound.  However, the statistical analysis and its description can be improved by the following.

First, the table of descriptive statistics need to include the sample means and standard deviations of the measured outcome variables.

==> Please see the red letters in 5 pages, 3.2. Part, Results.

And, please see the Table 3.

We authors added and described.

Second, you need to make clear that the regression models you estimate and report are conventional multiple regression models with the (Likert) adaptation and happiness scales as the outcome variables.  And you need to cite statistical methodological research  or prior studies that conclude that such multi-item Likert scales can be treated as interval scales (for a conventional multiple regression analysis), that is, that such regression analyses generally lead to inferences that are similar to those that treat the scales as ordered categorical scales with intervals between integers that are variable.

==> Please see the red letters in 4-5 pages, 2.4. Statistical Analysis.

We authors added and described.

Third, on first use in the text, you need to spell out the names of the model fit statistics in the text on first use, e.g., GFI, AGFI, etc.  I know their meaning, but this is a general social science journal intended for readers who may not be statisticians.  Among these model fit statistics, you also need to list R-squared,

==> Please see the red letters in 5-6 pages, 3.3. part, Results.

We authors added them.

Fourth, in Figure 2, you report the regression model estimates for the  "modified final model".  Okay, but it would be good also to report first the path diagram of the estimated conceptual model -- without the link from Adaptation to Life Satisfaction and then report the estimates for this final model, thus showing the model contribution of the Adaptation to Life Satisfaction link.

==> Please see the red letters on the last line in 2 pages, Introduction.

And please see the Figure 2.

We authors added and described.

Fifth, Table 3 needs work:  You need to make clear which of the coefficients relate to which outcome variable.

==> Please see the red line in Table 4.

We authors made an effort to amend.

Reviewer 2 Report

To:

Editorial Board

Title: “Relation Modeling of Factors Influencing Life Satisfaction and Adaptation of Korean Older Adults in Long-Term Care Facilities”

Dear Editor,

I read this manuscript and I think that:

Please revise the English of the paper due to typos. Inclusion and exclusion criteria should be better described. A flow chart of the study should be provided. The authors should include the number of pharmacological treatments of the patients. Please update table 1. A multivariate regression analysis should be performed in order to evaluate the role of confounding factors on results. The role of care manager should be discussed. Please consider the paper from Ciccone MM et al. Vasc Health Risk Manag. 2010 May 6;6:297-305.

Author Response

Revision Comments

Reviewer 2

Comments and Suggestions for Authors

Dear Editor,

I read this manuscript and I think that:

Please revise the English of the paper due to typos.

==> Please see the red letters throughout manuscript.

We authors made constantly effort to amend.

Inclusion and exclusion criteria should be better described.

==> Please see the red letters in 3 pages, 2.1. part, Material and Methods.

We authors added and described.

A flow chart of the study should be provided.

==> Please see the Figure 3.

We authors added.

The authors should include the number of pharmacological treatments of the patients.

==> Please see the red letters in 3 pages, 2.1. part, Material and Methods.

We authors added and described.

Please update table 1. A multivariate regression analysis should be performed in order to evaluate the role of confounding factors on results.

==> Please see the red letters in 4-5 pages, 2.4. part, Material and Methods.

We authors added and described.

The role of care manager should be discussed. Please consider the paper from Ciccone MM et al. Vasc Health Risk Manag. 2010 May 6;6:297-305.

==> Please see the red letters in 7 pages, Discussion.

We authors added and described.

Round 2

Reviewer 1 Report

The revised version of this manuscript has been responsive to the comments on the previous version.  It still needs some work along the following lines.

In addition to including the estimated model in Figure 2 that does not contain the direct effect of Adaptation on Life satisfaction, you need to compare the goodness-of-fit of the full model in Figure 4 with that of the preliminary model in Figure 2, e.g., the R-squared statistic for the Life satisfaction endogenous variable in Figure 4 with the corresponding R-squared statistic in Figure 2, in order to show that the inclusion of the direct effect from the first endogenous variable (Adaptation) on the second endogenous variable (Life satisfaction) adds explanatory power to the model.  And you also need to comment on the relative magnitudes of the estimated regression coefficients in the two models and how they change when you include the direct effect of Adaptation on Life satisfaction in the model.  For example, there are substantial decreases in the regression coefficients for Friend support and Staff support as you go from the model of Figure 2 to that of Figure 4 -- indicating that a sizeable part of the effects of these exogenous variables on Life satisfaction is mediated by Adaptation.  This is an interesting finding and you should relate it to prior (published) research.

On p. 6 of the text, you state "Among the endogenous variables in this study, depression..."  Is depression an endogenous variable or an exogenous variables?  In the path diagrams, it is exogenous?  If these are partial path diagrams that are endogenous to a set of exogenous variables, you need to make this clear.

Author Response

Revision Comments

Reviewer 1

Comments and Suggestions for Authors

The revised version of this manuscript has been responsive to the comments on the previous version.  It still needs some work along the following lines.

In addition to including the estimated model in Figure 2 that does not contain the direct effect of Adaptation on Life satisfaction, you need to compare the goodness-of-fit of the full model in Figure 4 with that of the preliminary model in Figure 2, e.g., the R-squared statistic for the Life satisfaction endogenous variable in Figure 4 with the corresponding R-squared statistic in Figure 2, in order to show that the inclusion of the direct effect from the first endogenous variable (Adaptation) on the second endogenous variable (Life satisfaction) adds explanatory power to the model. 

==> Please see the red letters in 5-6 pages, 3.3 part, Results.

We authors amended and described them.

And you also need to comment on the relative magnitudes of the estimated regression coefficients in the two models and how they change when you include the direct effect of Adaptation on Life satisfaction in the model.  For example, there are substantial decreases in the regression coefficients for Friend support and Staff support as you go from the model of Figure 2 to that of Figure 4 -- indicating that a sizeable part of the effects of these exogenous variables on Life satisfaction is mediated by Adaptation.  This is an interesting finding and you should relate it to prior (published) research.

On p. 6 of the text, you state "Among the endogenous variables in this study, depression..."  Is depression an endogenous variable or an exogenous variables?  In the path diagrams, it is exogenous?  If these are partial path diagrams that are endogenous to a set of exogenous variables, you need to make this clear.

==> Please see the red letters in 6 pages, 3.3 part, Results.

We authors amended and described them.

Reviewer 2 Report

Dear Editor,

I read the revised version of the paper and I think that the authors well addressed my previous comments. The paper improved very much as well as its scientific soudness.

Author Response

Revision Comments

Reviewer 2

I read the revised version of the paper and I think that the authors well addressed my previous comments. The paper improved very much as well as its scientific soundness.

==>Thank you so much.

Round 3

Reviewer 1 Report

The revisions generally are helpful.  You still need to address the following.

First, in Table 4, you need to include the estimated regression coefficient for the direct effect (and its level of statistical significance) from Adaptation to Life Satisfaction.  This is what distinguishes the modified final model from the hypothetical model.  

Second, you report the R-squared statistics for these two models (0.541 and 0.726).  Okay, but you also need to calculate and report a statistical significance test of whether this increase in R-squared values is statistically significant or not.  This is a really important point.  Does the inclusion of the direct effect path from Adaptation to Life Satisfaction significantly improve the predictive accuracy of the path model for the final outcome variable (Life Satisfaction)?  Since your sample size is modest (N = 229), the level of statistical significance for the difference in the R-squares may not be high, but even significance of the difference in R-squares at the 0.10 level is good.  

Author Response

Revision Comments

The revisions generally are helpful.  You still need to address the following.

First, in Table 4, you need to include the estimated regression coefficient for the direct effect (and its level of statistical significance) from Adaptation to Life Satisfaction.  This is what distinguishes the modified final model from the hypothetical model.  

==> Please see the red letters in the Table 4.

We authors added and amended.

Second, you report the R-squared statistics for these two models (0.541 and 0.726).  Okay, but you also need to calculate and report a statistical significance test of whether this increase in R-squared values is statistically significant or not.  This is a really important point. 

==> Please see the red letters in 6 pages.

Also, please see the red letters in the Table 4.

We authors added and amended.

Does the inclusion of the direct effect path from Adaptation to Life Satisfaction significantly improve the predictive accuracy of the path model for the final outcome variable (Life Satisfaction)? 

==> Please see the red letters in 5 pages.

We authors added and described.

Since your sample size is modest (N = 229), the level of statistical significance for the difference in the R-squares may not be high, but even significance of the difference in R-squares at the 0.10 level is good.  

==> Please see the red letters in 6 pages.

We authors added and described.

Round 4

Reviewer 1 Report

The revisions to this paper again have been good.  

The one suggestion I have at this point is that the sentence added on p. 6:

Expanding on adjusted R2 at the p<.01 based on moderate’s
 sample size, this study was same result.

needs work.  I think you are saying that the increase in the R2 of the modified model as compared to that of the hypothetical model (0.726 - 0.541 = 0.185) is statistically significant at the .01 level -- indicative that the addition of the path from the Adaptation to the Life Satisfaction variables yields a substantial improvement in model goodness-of-fit.

Author Response

Revision Comments

The revisions to this paper again have been good.  

The one suggestion I have at this point is that the sentence added on p. 6:

Expanding on adjusted R2 at the p<.01 based on moderate’s sample size, this study was same result.

needs work. I think you are saying that the increase in the R2 of the modified model as compared to that of the hypothetical model (0.726 - 0.541 = 0.185) is statistically significant at the .01 level -- indicative that the addition of the path from the Adaptation to the Life Satisfaction variables yields a substantial improvement in model goodness-of-fit.

==> Please see the red letters in 6 pages.

We authors amended and described.
